# The Relationship between Mental Disorders and the COVID-19 Pandemic—Course, Risk Factors, and Potential Consequences

**DOI:** 10.3390/ijerph19159573

**Published:** 2022-08-04

**Authors:** Agnieszka Markiewicz-Gospodarek, Aleksandra Górska, Renata Markiewicz, Zuzanna Chilimoniuk, Marcin Czeczelewski, Jacek Baj, Ryszard Maciejewski, Jolanta Masiak

**Affiliations:** 1Department of Human Anatomy, Medical University of Lublin, 4 Jaczewskiego St., 20-090 Lublin, Poland; 2Department of Psychiatric Nursing, Medical University of Lublin, 18 Szkolna St., 20-124 Lublin, Poland; 3Student Scientific Group, Department of Family Medicine, Medical University of Lublin, 6a (SPSK1) Langiewicza St., 20-032 Lublin, Poland; 4Department of Forensic Medicine, Medical University of Lublin, 8b Jaczewskiego St., 20-090 Lublin, Poland; 5II Department of Psychiatry and Psychiatric Rehabilitation, Medical University of Lublin, 1 Głuska (SPSK Nr 1) St., 20-059 Lublin, Poland

**Keywords:** SARS-CoV-2, COVID-19, pandemic, mental disorders, schizophrenia, depression, bipolar disorder, health problems

## Abstract

In this review the authors discuss that COVID-19 has already had a direct impact on the physical health of many people and that it appears to have put at risk the mental health of large populations. In this review, we also discuss the relationship between mental disorders and the SARS-CoV-2 infection. We convey the disorders’ risk factors and the more serious mental disorder consequences of COVID-19. People with mental health disorders could be more susceptible to the emotional responses brought on by the COVID-19 epidemic. The COVID-19 pandemic may adversely influence the mental health of patients with already diagnosed mental disorders. For the aim of dealing better with the psychological problems of people afflicted by the COVID-19 pandemic, new psychological procedures are required.

## 1. Introduction

The COVID-19 pandemic puts at risk the health and life of people around the world due to the possibility of death and to the experience of an individual sense of danger. In many patients with a confirmed COVID-19 infection, psychopathological symptoms were observed [1]. While many patients experience mental disorder symptoms after hospitalization for COVID-19, most do not go on to develop a new mental disorder [2]. The pathogenesis of mental disorders appearing due to the COVID-19 pandemic may include biological and psychosocial factors [3]. The long-term effects of COVID-19 infection are still being discovered and among them is an increased possibility of being diagnosed with mental health disorders [4]. Several studies have demonstrated that SARS-CoV-2 can be found in the brain tissue [5]. The presence of the virus was observed in the amygdala and hippocampus, which are the brain areas responsible for regulating mood and emotions [6]. COVID-19 can be associated with the onset of symptoms, such as insomnia, depressive disorders, concentration disturbances, anxiety, and memory impairment [7,8,9,10,11,12,13]. Moreover, the COVID-19 pandemic may adversely impact patients with already diagnosed mental disorders [14]. In this review, we focused on the course and risk factors of COVID-19 infection with the coexistence of mental disorders in patients with an earlier diagnosis of mental disorders, such as depression, bipolar disorder, schizophrenia, dementia, and developmental disorders, including autism. 

## 2. The Risk Factors of COVID-19 Infection with the Coexistence of Mental Disorders 

COVID-19 grew into a global pandemic at a frighteningly rapid pace. As of 30 September 2020, over 33 million cases of infection have been identified, along with one million deaths worldwide [15]. As of 22 May 2021, this had worsened further, rising to 166 million confirmed cases of COVID-19 and 3 million deaths [15]. It is estimated that mental disorders affect approximately 20–25% of the adult population (450 million worldwide) [16]. The emergence of the pandemic in 2019 did not help in improving the mental state of many people and has made it worse due to various social risk factors and pandemic stressors, including distress, anxiety, traumatic stress, depression, grief, suicidality, substance use, and burnout [17,18,19,20]. In addition, individuals who already had a mental disorder prior to the pandemic may be [18] at higher risk of infection due to trouble assessing health information reliably, adhering to preventive behaviors, existing barriers to accessing health care, or being on wards during hospitalization, which increases the risk of infection, especially if the hospital has a COVID-19 ward. Moreover, regarding the difficulties resulting from the course of mental illnesses listed above, there are other risk factors, such as gender or age [21,22,23]. An additional predisposing factor is the presence of comorbidities, such as cardiovascular disease, cancer, chronic obstructive pulmonary disease, and immune disorders [24]. During SARS-CoV-2 infection, systemic symptoms characteristic of infection with the above virus occurs, such as fever, cough, diarrhea, and anosmia, which, depending on the severity of the disease, may have a greater or lesser impact on the patients’ prognosis [25]. Moreover, the situation of patients is not improved by traumatic situations related to admission to the hospital ward, social and physical isolation, and the death of other patients and/or family members [26].

The Indian Psychiatric Society found a 20% increase in mental disturbances during the coronavirus outbreak because of social distance, isolation, fear of infection, marital conflicts, and loss of income [27]. At such critical times, a psychiatrist/psychologist/mental health specialist plays a crucial role in helping people who are unable to cope with the current situation. Table 1 shows selected studies that support the above statements, considering different age groups in different selected countries.

**Table 1 ijerph-19-09573-t001:** Selected studies considering outcomes related to mental disorders.

Type of Study	Number of People in the Study	Results	Ref.
Survey conducted in China on people over the age of 18; researchers recruited participants through an online questionnaire by sending a link to local groups via the WeChat app	1593	8.3% experienced anxiety and fear related to the presence of COVID-19 14.6% had symptoms of depression	[28]
Cross-sectional survey among Chinese students aged 12–18 during the COVID-19 pandemic via an online questionnaire	8079	43.7% had depression 37.4% had anxiety 31.3% had mixed disorders (depression/anxiety)	[29]
A survey of undergraduate students in two large cities in southwest China using online versions of the PTSD Checklist Civilian Version and the 9-question Patient Health Questionnaires (PHQ-9)	2485	The prevalence of probable PTSD and depression was estimated at 2.7% and 9.0% respectively	[30]
Survey in a population of people over the age of 18 living in various provinces of Turkey through an online questionnaire	343	23.6% showed symptoms of depression 45.1% showed symptoms of anxiety	[31]
A study conducted as an online survey data from the nationally representative “Understanding America Study” of the U.S. adult population	6585	29% of respondents reported symptoms of depression/anxiety	[32]
Survey of undergraduate students in New Jersey through an online survey	641	46% students reported elevated psychological stress 30.3% experienced increased health anxiety	[33]
A cross-sectional survey of the Hispanic population via online survey which contained 80 questions	3480	18.7% of respondents showed depressive symptoms 21.6% reported anxiety symptoms 15.8% reported PTSD	[34]
A cross-sectional survey of the Italian population via online questionnaire on an online survey platform	2766	Depression: (a) 67.3% of respondents had an average level (b) 17% were in the high range (c) 15.4% were in the extremely high range Anxiety: (a) 81.3% of respondents had an average level (b) 7.2% were in the high range (c) 11.5% were in the extremely high range Stress: (a) 72.8% of respondents were in the average range (b) 14.6% were in the high range (c) 12.6% were in the extremely high range	[35]
A cross-sectional survey of the Iranian population via an online 15-question questionnaire conducted in 31 provinces of Iran	10,754	50.9% had mild-to-severe anxiety symptoms 31.8% had mild-to-average symptoms 19.1% had severe-to-very severe symptoms	[36]
A cross-sectional survey of the Denmark population via online survey which included the five-item WHO-5 well-being scale; the survey also contained six questions regarding the experienced level of anxiety/depression over the past 2 weeks	2458	25.4% showed depressive symptoms	[37]

While the negative impact of the pandemic related to isolation/quarantine and precarious medical or work situations is clear, ambiguities are beginning to emerge regarding the role of social media. Globally, social media is a significant problem, causing difficulties in interpersonal interactions, isolation from society, and difficulty in separating work from private life [38]. In contrast, there are recent reports demonstrating the positive effects of social media on mental health during the COVID-19 pandemic [39]. Social media can mitigate the mental health impact of the COVID-19 pandemic by providing and maintaining social support during times of physical distancing measures and providing health information, telemedicine, and online psychological counseling [40]. However, the positive impact of the use of social media also has its dark side—it can trigger negative emotions, especially when the information is false or circumstantial [41]. Among patients burdened with mental illness, this can contribute to a worsening of the condition [42].

## 3. COVID-19 and Depression

Depression is a chronic mental disorder associated with affective, cognitive, and high-risk suicidal behaviors [43,44,45]. Currently, it is estimated that 5.0% of adults are affected by a depressive disorder and approximately 280 million people suffer from it worldwide [46]. In addition, it is not only a leading cause of disability, but it is also expected to become the most common burden of disease by 2030 [47].

When it comes to the pathogenesis of major depressive episodes, one of the hypotheses indicates that a decrease in the monoamine neurotransmitters, such as serotonin, noradrenaline, and dopamine, may lead to depressive symptom development [48]. In addition to monoamine hypothesis, chronic stress together with genetic factors may result in an increased production of pro-inflammatory mediators [49]. Thus, inflammation is also considered to be one of the causes of depression. Increased inflammatory marker levels, such as IL-1β, IL-2, IL-6, TNF-α, and CRP, and a decrease in anti-inflammatory cytokines, including IL-4 and IL-10, were noted in patients with depression [50]. Interestingly, there was a higher incidence of depression among patients with inflammatory diseases, which could confirm the proinflammatory status in depression [51]. 

Importantly, inflammatory cytokines are associated with disturbances at different pathophysiologic levels that are significant to depression, including neurotransmitter metabolism, neuroendocrine function, and neural plasticity [52,53]. It is also worth mentioning that high levels of pro-inflammatory mediators may increase the permeability of the blood–brain barrier and, consequently, enhance neuroinflammation and promote the entry of microorganisms [54]. Furthermore, increased peripheral cytokine concentrations were shown to activate inflammatory signaling pathways, affect brain function, cause deregulation of the hypothalamus–pituitary–adrenal (HPA) axis, and change monoamine levels [55,56]. Tryptophan is a major precursor of serotonin that is needed for its synthesis. Inflammatory cytokines were demonstrated to cause a reduction in monoamine levels by increasing the metabolism of tryptophan to kynurenine (KYN) [57].

Interestingly, COVID-19 was shown to have a negative impact on mental health by triggering or enhancing preexisting mental conditions, including depression [12,58,59]. This phenomenon may be supported by demonstrated pathophysiological mechanisms of COVID-19 that include a major inflammatory response and a massive production of cytokines [60]. These factors, which are common for the pathogenesis of both mentioned diseases, may be responsible for activating mechanisms associated with depression development [61]. The neurochemical changes induced by SARS-CoV-2 in depression are presented in Figure 1.

In addition, isolation and loss of social relations may contribute to a decline in cognition and mood, and may increase cortisol concentrations and negatively affect immune functioning [62,63]. Interestingly, a survey study carried out among an Australian population showed that individuals with depressive disorder were more concerned about access to appropriate medical care and experienced higher levels of psychological distress [64]. As social distancing was one of the most visible responses to the COVID-19 pandemic, it can also be associated with an increased likelihood to develop depression, especially among older adults and adolescents [65,66,67]. Several studies reported a negative impact of the pandemic on the anxiety and depression levels among adolescents [12,68,69]. Furthermore, elder adolescents presented higher rates of depression symptoms, with female adolescents being at a higher risk for depression and anxiety during the pandemic [67,68]. Due to stay-at-home orders and school closures, high levels of social media use were observed, as teenagers sought alternative ways to socialize with their peers. However, greater social media use was associated with higher depressive symptoms [69,70]. This is consistent with previous studies, which highlighted the negative relationship between social media use and depression [71,72]. 

**Figure 1 ijerph-19-09573-f001:**
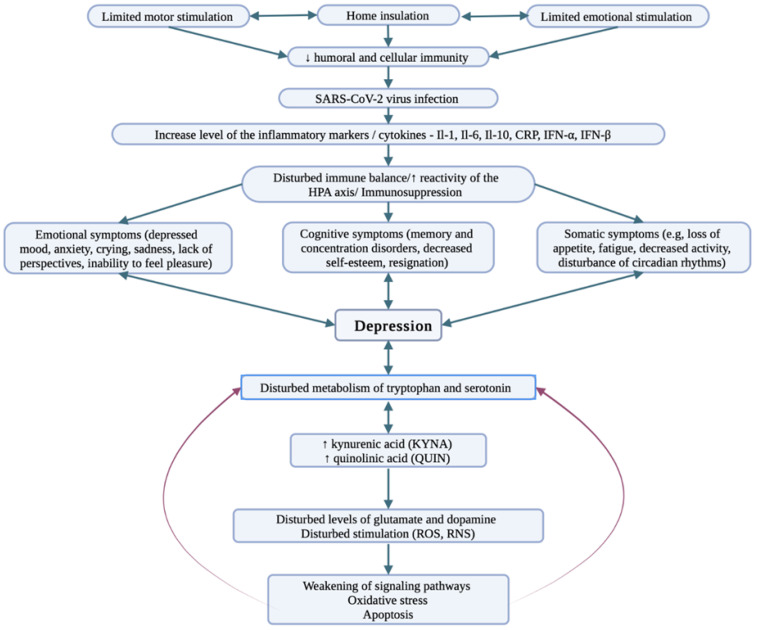
Depression—a pattern of neurochemical changes induced by SARS-CoV-2 [73,74,75]. Social isolation due to the possibility of SARS-CoV-2 virus infection decreases emotional and motor arousal resulting in decreased immunity. Consequently, the body is more susceptible to infections, including COVID-19 infection. When an infection occurs, inflammatory markers/cytokines increase in the body, leading to immunosuppression. Disturbed immune coordination causes discomfort to the entire body resulting in somatic, emotional, and cognitive symptoms, and the metabolism of neurotransmitters and their metabolites is impaired. The consequence is impairment of signaling pathways and development of depression.

## 4. COVID-19 and Bipolar Disorder

Bipolar disorder (BD) is a severe and chronic mental disorder characterized by recurrent episodes of mania, hypomania, and depression that vary in time [76]. Generally, BD affects more than 1% of the world’s population, with a distinction in prevalence between BD types I and II among males and females [76,77]. It is associated with functional deficits and neurocognitive dysfunction that includes aspects, such as attention, verbal learning, and memory [78]. Importantly, it was observed that patients with BD had the highest mean number of days out of role, which reflected the problems of disability and reduction in quality of life [79]. Furthermore, in comparison to other psychiatric conditions, BD was linked to the highest risk of suicide, with a suicide rate of approximately 15–20% among these patients [80,81]. Bipolar disorder is considered to be sensitive to factors that can disrupt biological and social rhythms. Previous studies have shown that environmental factors, such as life events, may affect the clinical course or trigger manic or depressive episodes [82,83,84]. While both negative and positive events were associated with depression and mania, positive events were more likely to evoke maniac episodes [85,86,87]. In addition, it was suggested that sleep disruption plays a role as a predictor of symptom exacerbation during BD [88]. Thus, preventive measures against COVID-19, such as social distancing or quarantine, could affect daily routines, sleep patterns, and social life, leading to an increased risk of manic and depressive relapses [89,90]. Interestingly, Carta et al. (2020) showed that rigid lockdowns in Caligari were related to a higher frequency of depressive episodes and biological rhythm modification in BD patients compared to patients under less restrictive measures in Tunis [59]. Biorhythm alterations in these Caligari patients resulted in sleep and social rhythm impairment. On the other hand, there is also a concern that during a manic or hypomanic episode, patients may not follow the social distancing restrictions, which could put them at a higher risk for infection [91]. 

A survey-based study from the initial wave of COVID-19 in Australia indicated that distress levels and depression were higher in individuals with a mood disorder, especially in BD patients [64]. Contrary to previous studies, male patients appeared to be more affected than the opposite gender during the COVID-19 pandemic [64,92,93]. The results also showed that the BD group was more concerned about unemployment or reduced employment risk compared to the control group and depressive disorder patients. The heightened economic burden caused by higher health-care costs for bipolar disorder patients could explain their concerns [64,94]. In addition, it was observed that working from home was more difficult and induced more stress among individuals with mood disorders [64]. Yocum et al. (2021) also showed that BD patients, compared to the control group, were more affected by home confinement and isolation restrictions. Firstly, these individuals presented high levels of distress, which were persistent or decreased with a much smaller magnitude, compared with health controls [95]. In Yocum et al.’s (2021) study, a slight decrease in distressed mood measurements was observed over time; however, the control group appeared to recover faster. In addition, during the stay-at-home period, no changes in quality of sleep were observed in the control group, whereas BD patients were more likely. However, BD patients were more likely to increase sleep duration [95]. 

Interestingly, Fellendorf’s (2021) study showed contradictory results, as BD individuals experienced worse subjective sleep quality, with reported sleep latency and daytime sleepiness [96]. In addition, this group of patients was also reported to gain weight, which was not compensated for, due to their decrease in physical activity compared to the control group [97]. Given all the burdens associated with BD, COVID-19 may pose a serious challenge to these patients.

## 5. COVID-19 and Schizophrenia

Schizophrenia is a disease, or rather a group of diseases, which is characterized by a varied picture and diverse clinical course, and it is the most serious and undoubtedly the most disabling mental illness [98]. The characteristic long-term course is manifested by the tendency to recurrence, protraction, or even perpetuation of disorders, often leading to a defensive lifestyle, characterized by withdrawal, passivity, and restriction of the expression of emotions, interests, and needs [99]. Research on schizophrenia shows that it is unevenly distributed in society [100]. It is estimated that its incidence is similar all over the world and amounts to about 1% of the world’s population [101]. The common feature is the occurrence of symptoms of disintegration (breakdown, splitting) of mental functions [98]. The background, i.e., the etiology of the disease, is unknown, while the disease itself manifests itself with psychotic symptoms that clearly impair functioning and include disorders of affect, thinking, and behavior. Schizophrenia is perceived as a serious, chronic mental illness [102]. 

Economic and ethnic differences appear to have a significant impact on the morbidity and treatment outcomes of COVID-19 [103]. Marginalized groups have increased COVID-19 morbidity and mortality [104], and individuals with severe mental illness may be disproportionately affected [105]. Indeed, individuals living with schizophrenia spectrum disorders (SSD) appear to be at higher risk for SARS-CoV-2 infection [106] due to their improper physical health, greater social impairment, reduced economic status, and shorter life expectancy and higher mortality, unrelated to the presence of infectious diseases, including COVID-19 [107]. Considering the above-mentioned behavioral complications that may have a significant impact on quality of life (including coping with daily activity, maintaining healthy (physical) behaviors, sense of satisfaction, level of aspirations, expectations, and meeting needs or social relations), schizophrenia can seriously complicate one’s life [108]. This is due to repeated periods of exacerbations and remissions, number of hospitalizations, increasing negative symptoms, or the need for long-term medication [109]. 

COVID-19 has already had a great impact on the general population worldwide and patients diagnosed with schizophrenia are at a greater risk of infection and its consequences, due to difficulties following preventive rules and the presence of other medical comorbidities [110]. To prevent potential infections among patients diagnosed with schizophrenia, some authors have conveyed that these patients should be well cared for. Additionally, some authors have suggested that physicians caring for COVID-19 patients should pay special attention to those with schizophrenia, as the latter may minimize their respiratory symptoms or have difficulty clarifying their symptoms [111]. According to these authors, the family also has an important role, as they should make the decision to contact their physician as soon as possible, in the traditional format, rather than via telemedicine, and limit access to media to avoid unnecessarily exposing patients diagnosed with schizophrenia to stressful situations if their symptoms worsen [112]. 

At least two studies in the USA have reported an increased frequency of COVID-19 among people with psychiatric disorders, including depression and schizophrenia [16,113]. A study conducted by Nemani et al. (2021), involving 26540 patients, including 7348 patients testing positive for SARS-CoV-2 (including 1% of patients with a history of schizophrenia spectrum disorder, 7.7% with mood disorders, and 4.9% with anxiety disorders), confirmed an increased risk of COVID-19 mortality among patients with schizophrenic disorders [114]. In contrast, mood disorders and anxiety disorders were not associated with COVID-19 mortality risk [115]. Another study, conducted in Israel, also confirmed that schizophrenia had a higher incidence of morbidity and mortality due to COVID-19 infection, compared to sex- and age-matched controls [115]. In another study, conducted in France, involving 50,750 patients hospitalized with COVID-19 from February to June 2020, there were several cases of patients with established schizophrenia (1.6%) [116]. Patients with schizophrenia had increased in-hospital mortality compared to controls (patients not diagnosed with severe mental illness) [117]. The results of this study indicate that there is a disparity in health and health care between patients with diagnosed schizophrenia and patients without such a diagnosis. The disparities are due to, among other things, differences based on the age or clinical profile of patients with a diagnosis of schizophrenia. 

## 6. Dementia in COVID-19 

The COVID-19 virus has become a particular threat to people with Alzheimer’s disease (AD) and other dementias. Clinical data suggest that even when considering other medical comorbidities, such as hypertension and diabetes, dementia is the essential independent risk factor for COVID-19 infection [118]. According to Tahira et al. (2021), dementia of all causes was associated with a significantly higher risk of COVID-19 diagnosis in in-patients 80 years of age and older (OR = 5.837, *p* < 0.001) [119]. The cognitive impairment and neuropsychiatric disorders make it difficult for people with dementia to understand and follow rapidly changing safety guidelines. Inability to comply with safety measures and specific dementia-associated symptoms, such as agitation or wandering, contribute to an increased risk of infection. Moreover, with the progression of the disease, patients become solely dependent on their caregivers [120]. While managing COVID-19 requires isolation and quarantine, this can hardly be achieved for this group of patients [121]. Physical distancing is not possible for people with dementia who cannot cope with basic activities of daily living. Therefore, the rates of infection are incomparably high in nursing or care homes where residents and limited staff gather in proximity [122,123]. At the peak of incidence in U.S. nursing homes, the COVID-19 rates exceeded as much as 20 cases per 1000 resident-weeks [124]. 

In addition to an increased risk of infection, patients with dementia are also at a higher risk of a severe course of the illness and a higher mortality [125,126]. The elder population, in which dementia typically develops, is prone to an unfavorable course of infection [127,128]. In a meta-analysis by Saragih et al. (2021), dementia was the main factor influencing poor health outcomes and high rates of mortality in older adults with COVID-19 infection (OR = 2.96, *p* = 0.224) [129]. Other dementia risk factors (obesity, cardiovascular disease, hypertension, and diabetes mellitus) are also risk factors for life-threatening respiratory failure and multi-organ damage during COVID-19 [130,131]. Importantly, dementia patients are more likely to develop these comorbidities compared to the population of the same age without dementia [132].

An additional mechanism that makes dementia patients more vulnerable to neurological complications from COVID-19 infection is their increased susceptibility to neuroinflammation. Apart from increasing the risk of Alzheimer’s disease, the ApoE ε4 genotype exacerbates microglia-mediated neuroinflammation [133,134]. A cohort study showed that the risk of hospitalization due to COVID-19 was 2.3- to 4.0-fold higher among homozygous for APOE ε4 than among patients with the common APOE ε3/ε3 genotype [135]. Moreover, Lim et al. reported multiple-times-elevated expression of the ACE2 gene in the brain tissues of patients with Alzheimer’s disease in comparison with individuals without the disease [136]. 

In addition to the immediate threat that the severe course of COVID-19 infection brings to the life of dementia patients, the prolonged restrictions have also had important neuropsychiatric consequences [137]. In a qualitative study by Smaling et al. (2022), health-care providers noticed cognitive decline as well as increased restlessness and aggression in people with dementia, due to restrictions on social contacts and activities [138]. In care homes, elderly people experienced distress due to the lack of visits from family members and limited interactions with fellow residents. Symptoms, such as depression, anxiety, agitation, and sleep disturbances, have been reported to increase during the periods of lockdown [139,140,141]. Patients with lower levels of cognition and those who lived alone were most at risk of developing negative neuropsychiatric effects during the lockdown [121,142]. Ismail et al. (2021) found that, in a study on a group of 36 patients, the monthly decrease in MMSE (mini-mental state examination) results before lockdown was 0.2 ± 0.1 points, while it was 0.53 ± 0.3 points during lockdown, which was statistically significant (*p* = 0.001) [143]. Moreover, once infected by COVID-19, there is a high probability that patients with dementia develop acute neuropsychological conditions, such as delirium [144]. Delirium superimposed on dementia is associated with accelerated cognitive and functional decline and mortality. Moreover, a diagnosis of delirium may be difficult because its manifestations, such as sleep disturbances, agitation, hallucinations, and aggression, resemble the features of dementia [145]. 

## 7. Developmental Disorders in COVID-19 

People with developmental disabilities appear to be at greater risk for severe outcomes from COVID-19 [146]. In a meta-analysis by Vai et al. (2021), the presence of developmental disorders was associated with an increased risk of COVID-19 mortality (OR = 1.73 (95% CI-1.29–2.31)) [24]. However, the COVID-19 pandemic has negative health outcomes in children beyond the effects of a viral infection. Panic and stress are the two main social consequences of the pandemic. Children with developmental disabilities are particularly vulnerable to those mental health effects in the period of social isolation and school closure [147]. It is estimated that in 2016, 52.9 million children under 5 years of age had developmental disabilities [148]. 

These children are more likely to develop mental health disorders, such as generalized anxiety disorder, obsessive-compulsive disorder (OCD), mood disorder, and psychotic disorder, compared to the general population [149,150]. Initial studies in populations without developmental disabilities have shown symptoms of posttraumatic stress disorder, the onset of psychotic symptoms, and an increase in suicidal thoughts and attempts related to fear caused by COVID-19 [151,152,153]. Therefore, such conditions can be anticipated in individuals with developmental disabilities, considering their higher risk for mental health problems [154].

One of the biggest stressors during the pandemic was the sudden disruption of people’s daily routines [155]. It is especially challenging for children with autism spectrum disorders (ASDs). Many ASD patients have difficulties in receptive communication skills and may experience information processing delays [156]. Studies have shown that scheduled activities reduce challenging behaviors among children with ASD [157]. After the pandemic outbreak, these children had difficulty understanding the rapidly changing daily activities, intensified by an inability to communicate effectively, and such a situation may elicit major anxiety, frustration, and emotional breakdowns [158,159].

Children with developmental disabilities rely heavily on therapy and support from specialized centers and schools. The COVID-19 pandemic caused a sudden suspension of rehabilitative and education services [160]. As consistency and intensity of intervention are key for children with developmental disabilities, this pause may increase the risks of social withdrawal and regressions of skills in these children. One of the most emphasized issues by parents of children with ASD was the failure of health-care facilities to meet these parents’ educational needs during the COVID-19 pandemic [161]. These children may also lack the cognitive ability to comprehend that their parents are acting as substitute teachers or therapists [162]. Moreover, a lack of therapists can contribute to a deterioration in mental health in these children and may lead to the development of psychiatric conditions. 

Apart from problems with access to education and psychological assistance, regular care from primary care settings was also limited due to the COVID-19 pandemic. The needs for children with developmental disabilities are greater because they are more likely to suffer from other medical conditions, including asthma, food and skin allergies, and headaches [163]. Children with developmental disabilities require direct interaction and building a relationship with their physician for the physician to perform an accurate diagnostic evaluation [147]. The use of telemedicine hardly allows a physician to have sufficient contact with these children [164]. Furthermore, telehealth may also be difficult to access for socioeconomically disadvantaged patients [164]. 

## 8. COVID-19 Vaccine and Mental Disorders

It was noted that people with mental disorders are more vulnerable to COVID-19 infection, worsening of a pre-existing psychiatric illness, and higher mortality [16,165,166,167]. COVID-19 vaccination uptake is crucial in high-risk groups and could bring significant health benefits [168]. Importantly, several countries have already prioritized vaccination for people with severe mental illness [169,170]. The relationship between COVID-19 vaccination and mental disorders was investigated in several studies [171,172,173].

Interestingly, studies regarding other infectious agents indicated that people with depression or depressive symptoms may present a reduced immune response to vaccination against influenza, measles, and Herpes Zoster, when compared to the controls. This phenomenon was especially common among older adults [174,175,176]. In addition, Russo et al. (1994) reported lower antibody responses to Hepatitis B vaccination in patients with schizophrenia, bipolar disorder, and depression [177]. According to the findings, which regarded other infectious agents, Mazereel et al. (2021) suggested that individuals with severe mental illness may have a reduced immune response to vaccination [171].

When it comes to psychotropic drugs and their potential interactions with the vaccines, Więdłocha et al.’s (2018) analysis showed a correlation between antidepressant treatment and an anti-inflammatory effect, as it affected the level of cytokines in patients with major depressive disorder [178]. On the other hand, antidepressant medication in elderly patients with major depression allows one to normalize the immune response to Herpes Zoster vaccination [175]. However, more research is needed in this area as there are no available data on how psychotropic drugs may affect the vaccine against COVID-19 response and its efficacy [171].

The literature is limited concerning the attitude of people with mental disorders towards vaccination against SARS-CoV-2 [172]. However, Curtis et al. (2022) indicated that lower vaccination coverage was observed among high-risk populations, such as people with severe mental illness [179]. It was supported by Bitan et al.’s (2021) studies, which showed that individuals with schizophrenia were less likely to receive vaccination for COVID-19 in Israel, especially among people aged 60 and above [180,181]. In addition, previous studies showed reduced rates for receipt of vaccination against influenza and pneumonia among patients with severe mental disorders [182]. Miles et al. (2020) highlighted several barriers to immunization, including lack of awareness, absence of medical recommendation, accessibility, and personal costs [183]. On the other hand, psychiatric patients in Chongqing (China), suffering from depression, presented high acceptance of COVID-10 vaccination and were more willing to pay for it when compared to healthy controls [184]. Moreover, Jefsen et al. (2021) observed high vaccine willingness in patients with mental illness and the general population in Denmark, with slightly lower rates in the first group [185].

Interestingly, it was suggested that vaccination may be effective in minimizing the differences in rates of hospitalization and mortality between the above-mentioned group of patients and the general population [181].

## 9. Long-Term Effects of SARS-CoV-2 and Their Potential Impact on the Further Course of Mental Disorders and the Convalescence Process

The medical and scientific community has made an enormous effort to fight the COVID-19 pandemic. However, the persistent health impacts of infection are yet to be investigated. During long-term follow-up, people who recovered from infection with other coronaviruses, such as MERS-CoV and SARS-CoV-1, developed long-term psychiatric conditions, including anxiety, depression, trauma, and sleep disturbance [186,187]. 

After the COVID-19 pandemic outbreak, extraordinary measures were introduced to slow down the spread of the virus. The closure of schools, restaurants, and other public places significantly reduced social interactions. Moreover, it led to an increase in unemployment and financial distress for many people. These factors, as well as the constant feeling of being at risk of virus infection, have significantly affected the wellbeing and mental health of the general population [188]. Data from 2020 highlighted an increase in the number of people experiencing mental health problems, compared to previous years [189,190]. 

Furthermore, after being diagnosed with COVID-19, patients have a double risk of a newly diagnosed psychiatric disorder [191]. Anxiety, depression, and sleep disturbances were the most common in the short term after infection [12]. The appearance of some mental health symptoms may be the result of SARS-CoV-2 infection. Autopsy of COVID-19 patients provides evidence that SARS-CoV-2 can cross the blood–brain barrier [192]. The secretion of interleukins, tumor necrosis factor, and nitric oxide as an immune response to the virus is linked to mood disorders and sleep disturbances [193]. However, long-term psychiatric symptoms of COVID-19 convalescents were consistent with the general population [194]. This suggests that mental health sequelae are due to psychosocial factors and not to a direct long-term effect of the virus. Consequently, all patients recovering from COVID-19 should be screened for any neuropsychiatric symptoms to ensure early diagnosis and intervention.

## 10. Conclusions

COVID-19 has had enormous impacts on mental health. This review found that people affected with SARS-CoV-2 have a high epidemiological burden of developing long-term psychiatric conditions, including anxiety, depression, trauma, sleep disturbance, suicidal behavior, and many more mental health problems. It should be crucial for the global mental health community to prevent the psychological consequences of COVID-19 in different population groups. This review provides an overview of research on the potential consequences of COVID-19 on depression, bipolar disorder, schizophrenia, dementia, and developmental disorders. We also focused on long-term effects of SARS-CoV-2 and their potential impact on the further course of their mental disorders. After being diagnosed with COVID-19, patients have a higher risk of a newly diagnosed mental disorder. The appearance of some mental health symptoms may be the result of SARS-CoV-2 infection. This phenomenon may be supported by established pathophysiological mechanisms of COVID-19 that include a major inflammatory response and a massive production of cytokines. These factors may be responsible for activating mechanisms associated with mental disorder development. However, in the opinions of the authors of this article, further research on the effects of SARS-CoV-2 and its potential impact on the severity and development of mental health disturbances regarding existing mental disorders is needed. Gaining a better understanding of the influence that the virus has on a psychological state would surely be useful for the attempts to mitigate the effects of COVID-19-related mental health consequences.

## Data Availability

Not applicable.

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
