# Peer review of "The Relationship between Mental Disorders and the COVID-19 Pandemic—Course, Risk Factors, and Potential Consequences"

_ijerph, 2022, doi:10.3390/ijerph19159573_

Round 1
Reviewer 1 Report
I appreciate the authors' diligence and substantial improvements on their article based upon my previous feedback. The following are my remaining requested edits and corrections, where once again I am using quotation marks to focus on what I am referring to.
--include citations/references after the statement in all the following lines: 35, 63, 144, 218, 239, 245, 359
--use a capital first letter after the colon in the references in all the following lines: 434, 496, 527, 534, 537, 575, 579, 595, 600, 603, 605, 611, 613, 625, 628, 633, 659, 667, 678, 681, 698, 710, 716, 733, 754, 756, 757, 760, 765, 768, 822, 826, 832
--use italics for the titles of internet articles in all the following lines: 458, 474, 509 & 510; 522, 524, 584, 590, 592, 649, 693,
--delete "et al." and include all the authors (up to first 20) in all the following lines: 567, 625, 628, 630, 716
--ue small first letters for all the main words (other than the first main word and first main word after a colon) in the titles of articles in all the following lines: 571, 639 & 640, 654, 744 & 745
--line 19: change phrase to "and that it appears to have put at risk the mental health of a large population"
--line 20: insert "we also" before "discuss"
--lines 21 & 22: change phrase to "the disorders' risk factors adn the more serious mental disorder consequencw of COVID-19. People with"
--line 24: change "With" to "For'
--line 25: changae "for" to "with" afater "better"
--line 66: put comma after "which"
--line 97: insert "a" before "depressive"
--line 103: insert "the" before "monoamine'
--line 120: change "to" to "for" after "needed"
--line 137: insert "the" before "COVID-19"
--line 171: insert "events" after "positive"
--line 172: change "symptoms" to "symptom"
--line 179: change phrase to "alteration in theswe Caligari patients resulted in sleep and social"
--line 190: change "the" to "their" before "concerns"
--line 195: change "et al." to "et al.'s"
--lines 198 & 199: chagne phrase to "whereas BD patients were more likely"
--line 200: change "and" to "as" before "BD"
--line 203: put comma after "for"
--line 214: change phrase to "about 1% of the world's population"
--line 216: put commas afater "background" and "disease"
--line 238: change phrase to with schizophrenia, some authors have conveyed that these patients should be well cared for"
--line 239: change phrase to "Additionally, some authors have suggestede that physicians caring"
--line 241: delete the extra space before "minimize" and insert "their" before "symptoms"
--line 242: change "The" to "According to these authors, the"
--line 243: put comma after "possible"
--line 244: insert "their" before "symptoms"
--line 254: change "with" to "that" after "confirmed"
--line 256: change phrase to "hospitalized with COVID-19 from"
--line 311: insert "in a study of" before "a group of"
--line 314: leave spaces before and after the equal sign
--line 319: put comma after "aggression"
--line 327: change "their" to "these" before "mental"
--line 345: insert "and" before "such a situation"
--line 350: put comma after "disabilities"
--line 355: change "substituting" to "substitute"
--line 358: delete "the" after "assistance,"
--line 361: delete comma after "food"
--line 363: change 'the" to "an" before "accurate"
--line 410: change "Further" to "However, in the opinions of the authors of this article, further"
--line 412: insert "is needed" after "disorders'
--line 413: insert "a" before "psychological"
--line 430: put comma after "Ser"
--line 759: correct spelling of "related"
Author Response
*Please see the attachment.
Dear Reviewer,
Thank you very much for reviewing our manuscript. We appreciate the interest and commitment you have provided for this work. We are very grateful for your extremely precious comments. We are convinced that thanks to your suggestions this manuscript will be much more valuable.
Reviewer 2 Report
Thank you for allowing me to read another version of this manuscript. I can definitely see some improvements – the grammar and quality of English is much improved and more references have indeed been provided. I appreciate the effort.
Unfortunately, however, in my option this paper still fundamentally falls short of being a useful contribution to the field at this point in time. Given that there are so many systematic reviews and other types of reviews on the same topic now, I find the justification of selecting papers simply based on the authors’ opinion and experience quite unconvincing and liable to omissions. Key studies have been missed, particularly those published in 2022. For example, they cite Bitan et al's older paper, but not their newer ones, which report on post-vaccination trends in covid mortality among people with mental illness.
Finally, the quality of writing and referencing still needs improvement. To be more specific and constructive in my feedback I have taken one section as an example below, but such comments are relevant to many sections of the manuscript.
Example section:
The Risk Factors of COVID-19 Infection with the Coexistence of Mental Disorders
Manuscript statement: “The emergence of the pandemic in 2019 has not helped improve the mental state of many people and has made it worse due to various factors.”
Comment – no references, vague statement.
Manuscript statement: “In addition, individuals who already had a mental disorder prior to the pandemic may be [16] at higher risk of infection due to trouble assessing health information reliably, adhering to preventive behaviors, existing barriers to accessing health care, or being on wards during hospitalization, which increases the risk of infection, especially if the hospital has a COVID ward.”
Comment – it is not clear from the positioning of reference 16 what point is being supported by this reference. That people with mental illness are at higher risk of infection? That they have trouble assessing health information reliably, adhering to preventive behaviors, or face existing barriers to accessing health care? Or that being on wards during hospitalization increases the risk of infection? Ref 16 refers to a study that used an online survey to assess anxiety and depressive symptoms in the general population during quarantine so it is not clear how this reference would support any of these statements.
Manuscript statement: "Moreover, in regard to the difficulties resulting from the course of mental illnesses listed above, there are other risk factors, such as male gender, obesity, and age."
Comment – no references, vague statement. Mixes modifiable factors (obesity) with non-modifiable demographic factors (age).
Manuscript statement: "Social distance, isolation, fear of infection, marital conflicts, and loss of income cause great distress and impact an already stressed health care system [20]."
Comment: Which healthcare system? What country/countries? It is too vague to imply that all healthcare systems globally, which vary hugely in their resources and organization - are ‘stressed’.
Manuscript statement: "The papers presented below were selected randomly considering the relevance of the research according to the authors’ opinion."
Comment: This is not the definition of ‘random’ as used in a research context. Selecting papers cannot be simultaneously 'random' and based on 'relevance'.
TABLE 1
Table 1 presents information haphazardly - it doesn’t provide sufficient level of detail to understand what type and standard of evidence these studies provide to be useful to the reader.
Taking the first line of Table 1 (ref 22) as an example:
Manuscript statement (Table 1): A study conducted in China on people over the age of 18
Comment: What kind of study? Cohort, survey, qual, quant, prospective, case-control? General population?
Manuscript statement (Table 1): 8.3% felt anxious and fearful
Comment: …anxious and fearful of what?
Manuscript statement (Table 1): 14.6% had symptoms of depression
Comment: When? After or before infection/hospitalisation? During isolation/lockdown?
Author Response
*Please see the attachment
Dear Reviewer,
Thank you very much for reviewing our manuscript. We appreciate the interest and commitment you have provided for this work. We are very grateful for your extremely precious comments. We are convinced that thanks to your suggestions this manuscript will be much more valuable.

Reviewer 3 Report
I appreciate the authors' effort to improve the article. However, this does not change my position on the originality of the topic raised
Author Response
* Please see the attachment.
Dear Reviewer,
Thank you very much for reviewing our manuscript. We appreciate the interest and commitment you have provided for this work. We are very grateful for your extremely precious comments. We are convinced that thanks to your suggestions this manuscript will be much more valuable.
This manuscript is a resubmission of an earlier submission. The following is a list of the peer review reports and author responses from that submission.
Round 1
Reviewer 1 Report
Thank you for the opportunity to review this paper about the impact of Covid-19 on mental health. It is a very well documented review, however, it need some changes in my opinion. In the abstract ''We related to the disorders risk factor and consequences of the COVID-19 for the mental disorders''(page 1 row 20) should be change, it is not clear what the authors wanted to say.
Also, after the 1. Introduction there is directly 2. The risk factors of COVID-19 infection with the coexistence of mental disorders, you should put a few paragraphs for Methodology like how the literature review was performed, how the studies were selected. See page 2, Table 1 row 78; How you have selected the studies.
Reviewer 2 Report
This is a very important and timely article and I very much would like to see it get published in this journal. However, I think there are extensive English language problems that need to get corrected before it gets published, a number of citations/references that should be added, and there are also some technical issues, using the format of the American Psychological Association, though this can be decided upon by the journal editor. Since I am very supportive of this article getting published, I am including complete line by line detailed feedback on all of the above for the authors’ benefit..

Reviewer 3 Report
This paper attempts to review relationships between mental disorders and COVID-19. This is an important area for study and the authors have set themselves an ambitious task. It is clear that they have read many papers and put in some effort to prepare the manuscript. However, I fear that there are several fundamental problems with this paper.
1. The authors present no details of their literature review methodology, so I am not sure what type of review this is intended to be (e.g. narrative, scoping, systematic etc). I am unable to see how the authors collated, curated and evaluated this literature, which is crucial to establish its rigour and understand what gaps in evidence remain in the field.
2. The aims and objectives of the review are unclear. This area of work has generated a vast amount of literature in a short space of time, so the authors need to be clear about what they did/ did not intend to address.
3. The English and quality of writing does not meet the standard I would expect for a publishable article. It is not just that there are typos and grammatical errors; rather, it the standard and use of English makes it difficult to read and understand exactly what was done and follow the logic of the arguments presented.
4. References are presented sporadically, or not provided at all. There are many examples of this. For example, in the introductory section, the authors state:
"While many patients experience mental disorder symptoms after hospitalization for COVID-19, most do not go on to develop a new mental disorder".
Statements such as these should always be backed up by at least one reference, especially in a review article.
Furthermore, from my own knowledge of the field I can see that key references are missing. For example, there have been at least 4 systematic reviews of the relationship between COVID-19 and mental disorder. This review cites just one of them (Vai et al, 2021).
Reviewer 4 Report
The title of the article promised very interesting content, concerning – as I thought - a broad review of research on the consequences of the COVID-19 pandemic in people with mental disorders. These types of review papers, based on meta-analyses, are particularly valuable for researchers since they can get a compendium of information in a single article. Additionally, the analyses that would look at the association between COVID and pre-existing mental disorders seemed extremely interesting. In the article submitted for review, the authors have indeed provided an overview of the research on the topic, but it is pretty vague. Moreover, it discusses the groups of people with various mental health problems. It seems to me that an overview of information on such a large diversity of illnesses (seniors with dementia, children with ASD, people with depression, bipolar disorder, schizophrenia) is not the best idea. I have got the impression that the information is too superficial. Unfortunately, the author(s) only signal problems but do not explain them. I would recommend the article for students since the knowledge presented in a "nutshell" may be of help in the didactic process. It does not, however, seem to be a significant source of knowledge for researchers.